# A Convolutional Auto-Encoder for Haplotype Assembly and Viral Quasispecies Reconstruction

**Ziqi Ke and Haris Vikalo**
Department of Electrical and Computer Engineering
The University of Texas at Austin
ziqike@utexas.edu, hvikalo@ece.utexas.edu

## Abstract

Haplotype assembly and viral quasispecies reconstruction are challenging tasks concerned with analysis of genomic mixtures using sequencing data. High-throughput sequencing technologies generate enormous amounts of short fragments (reads) which essentially oversample components of a mixture; the representation redundancy enables reconstruction of the components (haplotypes, viral strains). The reconstruction problem, known to be NP-hard, boils down to grouping together reads originating from the same component in a mixture. Existing methods struggle to solve this problem with required level of accuracy and low runtimes; the problem is becoming increasingly more challenging as the number and length of the components increase. This paper proposes a read clustering method based on a convolutional auto-encoder designed to first project sequenced fragments to a low-dimensional space and then estimate the probability of the read origin using learned embedded features. The components are reconstructed by finding consensus sequences that agglomerate reads from the same origin. Mini-batch stochastic gradient descent and dimension reduction of reads allow the proposed method to efficiently deal with massive numbers of long reads. Experiments on simulated, semi-experimental and experimental data demonstrate the ability of the proposed method to accurately reconstruct haplotypes and viral quasispecies, often demonstrating superior performance compared to state-of-the-art methods. Source codes are available at `https://github.com/WuLoli/CAECseq`.

## 1   Introduction

Genetic material in living cells and viruses experiences mutations which lead to unique blueprints and/or may create potentially diverse and complex genomic communities. In humans, genetic mutations impact an individual's health by causing genetic diseases and rendering the individual predisposed to complex diseases. In general, genetic material of eukaryotic organisms is organized in chromosomes, each with two or more copies present in a cell; variations between chromosomal copies have major implications on cellular functions. Beyond the living organisms, genetic variations also occur in viruses where they lead to emergence of rich viral populations that co-exist as the so-called quasispecies; spectrum of such quasispecies is reflective of the proliferative advantage that particular mutations may provide to viral strains present in the community. Therefore, inferring the composition and studying evolution of genomic communities that emerge due to occurrence and accumulation of mutations provide valuable information about genetic signatures of diseases, and generally suggest directions for medical and pharmaceutical research. High-throughput sequencing technologies enable sampling of such genomic communities/mixtures (Schwartz, 2010; Clark, 2004; Sabeti et al., 2002); however, the composition inference is computationally challenging and of limited accuracy due to sequencing errors and relatively short lengths of sequencing reads.

**Haplotypes.** Perhaps the simplest manifestation of genetic diversity in an individual's genome are the variations between copies of autosomal chromosomes inherited from the individual's parents. An ordered list of single nucleotide polymorphisms (SNPs) on the individual's autosomal chromosomes is referred to as haplotype (Schwartz, 2010). While humans are diploids, i.e., have a genome organized in chromosomal pairs, many organisms are polyploids and thus their chromosomes (consequently, haplotypes) come in triplets, quadruplets and so on. Accurate assembly of haplotypes requires deep sequencing coverage, especially in the case of polyploids. Presence of sequencing errors and relatively short length of reads compared to distance between mutations, render haplotype assembly challenging (Motazedi et al., 2018).

**Viral quasispecies.** An even more challenging problem than haplotype assembly is the reconstruction of viral populations. RNA viruses such as HIV, HCV, Zika, coronavirus and so on, typically exist as communities of closely related yet still decidedly distinct variants present at different abundances (i.e., have varied relative frequencies). A collection of such variants (i.e., strains) is referred to as a viral quasispecies; studies of quasispecies are essential for the understanding of viral dynamics, including insight into processes that enable viruses to become resistant to drugs and vaccines. In addition to the challenges encountered in solving haplotype assembly problems, viral quasispecies reconstruction (for convenience also referred to as viral haplotype reconstruction) is even more challenging due to unknown viral population size and imbalanced abundances of viral strains.

## 1.1 Contributions

In this paper we propose CAECseq, a novel convolutional auto-encoder with a clustering layer, inspired by (Guo et al., 2017), for solving both haplotype assembly and viral quasispecies reconstruction problems. Auto-encoders are neural networks that can be trained to automatically extract salient low-dimensional representations of high-dimensional data in an unsupervised manner (Goodfellow, Bengio, and Courville, 2016). In auto-encoders, an encoder aims to compress input data to obtain useful feature embeddings while a decoder aims to reconstruct input data from the learned feature. The learned features have been proved to perform well in many fields including anomaly detection (Zhou and Paffenroth, 2017), image clustering (Guo et al., 2017), natural language processing (Socher et al., 2011), information retrieval (Kipf and Welling, 2016) and so on.

CAECseq's encoder consists of convolutional layers followed by a dense layer and converts reads into learned low-dimensional feature embeddings, while the decoder consists of a dense layer followed by deconvolutional layers and reconstructs the reads. After pre-training the convolutional auto-encoder to project reads to a stable low-dimensional feature space, we utilize k-means on the learned features to initialize parameters of the clustering layer. The convolutional auto-encoder and the clustering layer are then trained simultaneously to cluster reads without distorting the low-dimensional feature space, where the clustering task is guided by a target distribution aimed to reduce the MEC score.

Our main contributions are summarized as follows:

- We developed a convolutional auto-encoder with a clustering layer, CAECseq, for haplotype assembly and viral quasispecies reconstruction; CAECseq is trained to automatically group together reads originating from the same genomic component, processing sequencing data in an end-to-end manner. The ability of convolutional layers to capture spatial relationship between SNPs enables the proposed method to distinguish reads obtained from highly similar genomic components.

- Our proposed framework pursues indirect optimization of the MEC score, which enables use of mini-batch stochastic gradient; this, combined with dimension reduction of the reads, allows us to efficiently deal with massive amounts of long reads.

- We conducted extensive experiments on simulated, semi-experimental and experimental data, obtaining results which demonstrate the ability of the proposed method to efficiently and with high accuracy assemble haplotypes and reconstruct viral quasispecies from high-throughput sequencing data.

## 1.2 Related work

Majority of existing methods for haplotype assembly either directly or indirectly rely on partitioning reads into clusters according to their chromosomal origins. While early methods explored a variety of

metrics including minimum single nucleotide polymorphism (SNP) removal (Lancia et al., 2001) and maximum fragments cut (Duitama et al., 2010), the vast majority of more recent techniques is focused on minimum error correction (MEC) optimization (Lippert et al., 2002), i.e. determining the smallest number of inconsistencies between reads and the reconstructed haplotypes (Ke and Vikalo, 2020; Edge, Bafna, and Bansal, 2017; Xie et al., 2016; Bonizzoni et al., 2016; Patterson et al., 2015; Pisanti et al., 2015; Kuleshov, 2014). Existing MEC score optimization methods can be divided into two categories: those in pursuit of exact solutions to the MEC optimization problem, and computationally efficient heuristics. The former include (Wang et al., 2005), a method based on branch-and-bound integer least-squares optimization; (Chen, Deng, and Wang, 2013), a method based on integer linear programming, and (Kuleshov, 2014), a framework based on dynamic programming, which leads to very high computational complexity. The latter include (Levy et al., 2007), a greedy heuristic; HapCUT (Bansal et al., 2008), a max-cut algorithm; (Duitama et al., 2011), a greedy max-cut method; methods calculating the posterior joint probability of SNPs in a haplotype based on MCMC (Bansal et al., 2008) and Gibbs sampling (Kim, Waterman, and Li, 2007); HapCompass (Aguiar and Istrail, 2012), an approach based on flow-graphs; SDhaP (Das and Vikalo, 2015), a framework based on convex optimization; BP (Puljiz and Vikalo, 2016), an algorithm motivated by communication theory; and HapCUT2 (Edge, Bafna, and Bansal, 2017), a maximum-likelihood-based algorithm which is an upgraded version of HapCUT, to name a few.

Reconstructing polyploid haplotypes is more difficult than solving the same task for diploids due to the expanded search space. Existing methods capable of handling haplotype assembly of both diploids and polyploids include HapCompass (Aguiar and Istrail, 2012); HapTree (Berger et al., 2014), a Bayesian method; SDhaP (Das and Vikalo, 2015); BP (Puljiz and Vikalo, 2016); matrix factorization frameworks including (Cai, Sanghavi, and Vikalo, 2016) and AltHap (Hashemi, Zhu, and Vikalo, 2018); and GAEseq (Ke and Vikalo, 2020), a method based on a graph auto-encoder.

Finally, prior work on viral quasispecies reconstruction includes ViSpA (Astrovskaya et al., 2011), a method based on read clustering; ShoRAH (Zagordi et al., 2011), a method based on read-graph path search; QuRe (Prosperi and Salemi, 2012), an algorithm that relies on combinatorial optimization; QuasiRecomb (Töpfer et al., 2013), a technique based on a hidden Markov mode; PredictHaplo (Prabhakaran et al., 2014), an algorithm that relies on Dirichlet process generative models; aBayesQR (Ahn and Vikalo, 2017), an approach based on hierarchical clustering and Bayesian inference; TenSQR (Ahn, Ke, and Vikalo, 2018), a successive clustering framework using tensor factorization; and GAEseq (Ke and Vikalo, 2020), a graph euto-encoder technique. Among all the existing methods, GAEseq (Ke and Vikalo, 2020) is the only one designed to handle both haplotype assembly and viral quasispecise reconstruction problems. Note, however, that due to aiming to minimize the MEC score directly, GAEseq uses full-batch gradient descent which makes it exceedingly slow and practically infeasible when dealing with large numbers of reads.

## 2 Methods

### 2.1 Problem formulation

High-throughput sequencing platforms provide (possibly erroneous) reads that oversample a mixture of genomic components. Reads are much shorter than the sampled genomic components; their relative positions can be determined via mapping to a known reference genome. Figure 1 shows an example of the end-to-end (viral) haplotype reconstruction from sequencing data. Since the reconstruction task is focused on determining the order of heterozygous genomic sites, we only keep the single nucleotide polymorphisms (SNPs) and represent the informative data by an $n \times l$ SNP fragment matrix $S$ where $n$ is the number of reads and $l$ is the length of the haplotypes. After implementing read clustering to group together reads originating from the same genomic component, the reconstruction of haplotypes is enabled by determining the consensus sequence for each cluster. The reconstructed haplotypes form a $k \times l$ haplotype matrix $H$ where $k$ denotes the number of haplotypes. Our proposed method for read clustering is based on a convolutional auto-encoder with a clustering layer. Instead of clustering reads in the original space, which is done by the vast majority of existing methods and relies on the Hamming distance measure, we first project the reads to a low-dimensional space while maintaining the spatial relationships between SNPs by using the convolutional auto-encoder. Note that when calculating Hamming distance between two strains in the original space, SNPs are assumed to be independent of each other and therefore their spatial relationships are not taken into account. After learning the feature embeddings of reads, the reads are grouped using the clustering layer which

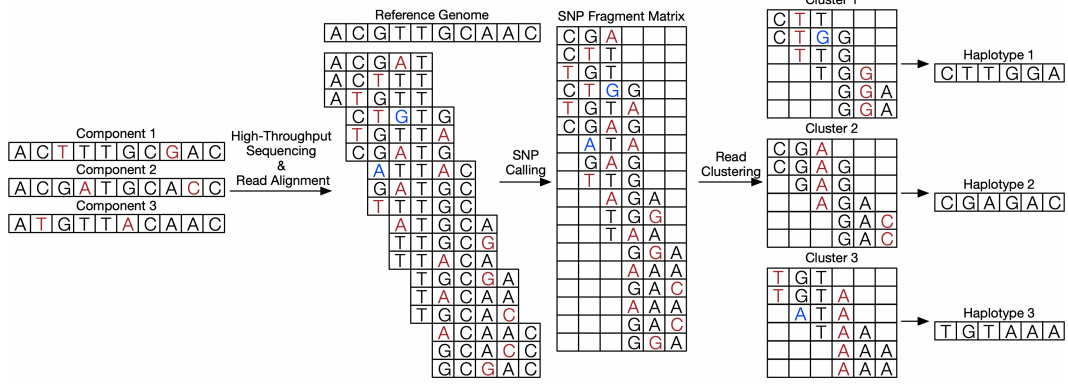

Figure 1: An example of the end-to-end (viral) haplotype reconstruction from sequencing data. An empty entry in a row indicates that the site corresponding to a column is not covered by the read corresponding to the row. SNPs are marked in red, sequencing errors are marked in blue.

estimates the probability of the reads' origin. Figure 2 illustrates the architecture of the proposed algorithm for read clustering. The reads are first one-hot encoded before being fed into the neural network, i.e., the four types of nucleotides are represented as $(1, 0, 0, 0)$, $(0, 1, 0, 0)$, $(0, 0, 1, 0)$ and $(0, 0, 0, 1)$, while the genome positions not covered by a read are represented as $(0, 0, 0, 0)$. The

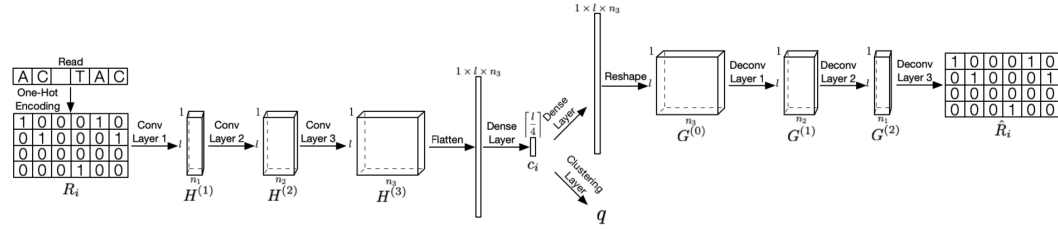

Figure 2: Architecture of the proposed algorithm for read clustering. The reads are first one-hot encoded and then fed into the neural network. Here $n_1$, $n_2$ and $n_3$ denote the filter size of the convolutional and convolutional transpose layers, and $q$ is the estimated probability of the read origin.

number of genomic components $k$ in haplotype assembly is known beforehand, but the population size of a viral quasispecies needs to be estimated. We follow the same strategy as in (Ahn and Vikalo, 2017) to automatically infer the population size based on the MEC improvement rate.

The performance of haplotype assembly is expressed in terms of the minimum error correction (MEC) score (Lippert et al., 2002) and correct phasing rate (CPR), also known as the reconstruction rate. Note that CPR can only be evaluated when the ground truth is available, which is typically not the case in practice. The MEC score is defined as the smallest number of inconsistencies between reads and the reconstructed haplotypes at positions that are covered by the reads, i.e.,

$$\text{MEC} = \sum_{i=1}^{n} \min_{j=1,2,\ldots,k} \text{HD}(S_i, H_j), \tag{1}$$

where $\text{HD}(\cdot, \cdot)$ represents the Hamming distance calculated only at the positions covered by the reads, $S_i$ is the $i^{th}$ SNP fragment and $H_j$ is the $j^{th}$ haplotype. CPR, essentially the average proportion of genetic variations that are perfectly reconstructed, is formally defined as

$$\text{CPR} = 1 - \frac{1}{kl}(\min \sum_{i=1}^{k} \text{HD}(H_i, \mathcal{M}(H_i))), \tag{2}$$

where $\mathcal{M}$ is the best one-to-one mapping from the reconstructed haplotypes to true haplotypes (Hashemi, Zhu, and Vikalo, 2018).

## 2.2 Convolutional auto-encoder

The convolutional auto-encoder consists of two symmetric parts: an encoder, composed of three convolutional layers followed by a dense layer for converting one-hot encoded reads into short feature embeddings; and a decoder, composed of a dense layer followed by three convolutional transpose layers for reconstructing one-hot encoded reads from the learned features. The operations of convolutional encoder (see Figure 2) can be formalized as

$$H^{(0)} = R_i \tag{3}$$

$$H^{(l)} = \sigma(H^{(l-1)} * W_{l-1}^{conv} + B_{l-1}^{conv}), \ l \in \{1, 2, 3\} \tag{4}$$

$$c_i = W_1^{dense} \cdot \text{Flatten}(H^{(3)}) + B_1^{dense}, \tag{5}$$

where $R_i$ represents the $i^{th}$ one-hot encoded read and $c_i$ denotes the low-dimensional representation of the $i^{th}$ read. $W$ and $B$ are weights and biases, respectively, $'*'$ is the convolution operator and $\sigma$ denotes the parametric rectified linear unit (PReLU) activation function (He et al., 2015). The deconvolutional decoder (see Figure 2) can be represented as

$$G^{(0)} = \text{Reshape}(\sigma(W_2^{dense} \cdot c_i + B_2^{dense})) \tag{6}$$

$$G^{(l)} = \sigma(G^{(l-1)} * W_{l-1}^{deconv} + B_{l-1}^{deconv}), l \in \{1, 2, 3\} \tag{7}$$

$$\hat{R}_i = G^{(3)} \tag{8}$$

where $\hat{R}_i$ is the reconstructed $i^{th}$ one-hot encoded read. Note that the learned features are restricted to be shorter than haplotypes to avoid learning useless features, and that the pooling layers are not utilized in order to better maintain the spatial relationships between SNPs. The convolutional auto-encoder can be trained by minimizing the reconstruction loss as ($\| \cdot \|_F$ is the Frobenius norm)

$$L_r = \frac{1}{n} \sum_{i=1}^{n} \|\hat{R}_i - R_i\|_F^2. \tag{9}$$

## 2.3 Clustering layer

Learnable parameters of the clustering layer are set to be the cluster centroids $\{\mu_j\}_1^k$, and thus the clustering layer is able to automatically group the learned feature embeddings into $k$ clusters based on the Euclidean distances between the feature embeddings and cluster centroids (Guo et al., 2017). The output of the clustering layer is an estimate of the probability of the read origins,

$$q_{ij} = \frac{(1 + \|c_i - \mu_j\|_2^2)^{-1}}{\sum_j (1 + \|c_i - \mu_j\|_2^2)^{-1}}, \tag{10}$$

where $c_i$ denotes the learned feature of the $i^{th}$ read, $\mu_j$ is the center of the $j^{th}$ cluster, and $q_{ij}$ denotes the probability that the $i^{th}$ read is from the $j^{th}$ genomic component. In order to facilitate the haplotype reconstruction task, we carefully design a target distribution by aiming to indirectly minimize the MEC score (direct MEC optimization requires full-batch gradient descent which hinders the ability of handling massive amounts of reads (Ke and Vikalo, 2020)). We follow 4 steps to acquire the target distribution: 1. Determine the origin of reads $I_i = \arg\max_j q_{ij}$. 2. Find the consensus strain $H_j$ in each cluster via majority voting. 3. Calculate $D_{ij}$, the Hamming distance between the $i^{th}$ read and the $j^{th}$ cluster. 4. Set $p_{ij}$ to 1 if $j = \arg\min_j D_{ij}$, and to 0 otherwise. In other words, $p_i$ is one of the $k$-dimensional standard unit vectors, with 1 in the $j^{th}$ position and the remaining entries 0. The clustering loss in the form of Kullback–Leibler (KL) divergence is given by

$$L_c = \sum_i \sum_j p_{ij} \log \frac{p_{ij}}{q_{ij}} \tag{11}$$

## 2.4 Optimization and post-processing

The convolutional auto-encoder (AE) is first trained by minimizing the reconstruction loss $l_r$ in order to reach a stable feature space. Next, the learnable parameters of the clustering layer are initialized using the cluster centroids acquired by implementing $k$-means on the low-dimensional learned features

of reads. The reconstruction loss $l_r$ and the clustering loss $l_c$ are then optimized simultaneously so that the clustering task helps lead to a feature space beneficial for MEC minimization. The combined loss function is defined as $L = (1 - \gamma)L_r + \gamma L_c$, where the parameter $\gamma$ is used to balance $L_r$ and $L_c$. Note that a large $\gamma$ distorts the feature space by reducing the effect of the convolutional AE while a small $\gamma$ leads to a feature space that is less helpful to the haplotype reconstruction task. Also note that $p_{ij}$ (treated as the ground truth of the origin of the reads) is updated only at each epoch instead of each iteration using a mini-batch of the reads. The MEC score is evaluated every time we update $p_{ij}$ and the training is terminated if the MEC score at the current epoch matches the previous one.

Since the proposed method does not minimize MEC score directly, a post-processing steps is used. In particular, after completing the training and reconstructing the haplotypes, we repeat steps 2, 3 and 4 to update the reconstructed haplotypes until the MEC score converges (Ahn, Ke, and Vikalo, 2018).

### 2.5 Hyper-parameters settings and computational platforms

The hyper-parameters of CAECseq are determined by training on five simulated tetraploid datasets with sequencing coverage $20\times$, and are validated on different five simulated tetraploid datasets with the same coverage. The reported results were obtained on test data. The algorithm is run 10 times, each time with a random initializations of the neural network, and the model achieving the lowest MEC score is selected. For all the experiment results, $\gamma = 0.1$, the batch size is $\left\lceil \frac{n}{20} \right\rceil$, the learning rate is set to 0.001, the number of pre-training epoch is 100, the length of learned features is $\left\lceil \frac{l}{4} \right\rceil$. Moreover, $n_1$, $n_2$ and $n_3$ are set to 32, 64 and 128, respectively. The kernel sizes of the convolutional layers are $(4, 5)$, $(1, 5)$ and $(1, 3)$, and those of the convolutional transpose layers are $(1, 3)$, $(1, 5)$ and $(4, 5)$. All the strides are set to 1. Training and testing is done on a machine with a 3.70GHz Intel i7-8700K processor, 2 NVIDIA GeForce GTX 1080Ti computer graphics cards and 32GB RAM.

## 3 Results

### 3.1 Performance comparison on semi-experimental *Solanum Tuberosum* data

The performance of CAECseq is first tested on semi-experimental *Solanum Tuberosum* ($k = 4$) data and compared with state-of-the-art methods including HapCompass (Aguiar and Istrail, 2012), a method based on graph theory; H-PoP (Xie et al., 2016), an algorithm utilizing dynamic programming; AltHap (Hashemi, Zhu, and Vikalo, 2018), an algorithm using matrix factorization; and GAEseq (Ke and Vikalo, 2020), a framework based on a graph auto-encoder. The semi-experimental data is created by selecting a reference genome, simulating mutations to generate haplotypes, generating reads with shotgun sequencing, aligning reads to the reference genome, and, finally, calling SNPs. The reference genome 5000 bp long is randomly selected from *Solanum Tuberosum* chromosome 5 (Potato Genome Sequencing Consortium, 2011). Haplotypes are then synthesized using *Haplogenerator* (Motazedi et al., 2018) which generates independent mutations on a genome according to a log-normal distribution. Following (Motazedi et al., 2018), the mean distance between mutations and the standard deviation are set to 21 bp and 27 bp, respectively, resulting in haplotypes of length about 150. Illumina's MiSeq reads of length $2 \times 250$ bp with mean inner distance 50 bp and standard deviation 10 bp are generated utilizing *ART* (Huang et al., 2012), where the sequencing error rate is automatically inferred by this tool from the data. Read alignment is performed using *BWA-MEM* (Li and Durbin, 2009), where the reads with mapping scores lower than 40 are filtered out for quality control. SNP positions are determined by comparing the frequency of the alternative allele at any given site with a predefined threshold. Sequencing coverage is again varied from $5\times$ to $40\times$ with step size $5\times$, resulting in read numbers that range from approximately 200 to 1600. For each coverage setting, 10 data samples are generated and processed to evaluate the mean and standard deviation of the MEC scores and CPR achieved by CAECseq and the selected competing methods. Table 1 compares the performance of CAECseq and the competing methods in terms of MEC scores and CPR for datasets with sequencing coverage $5\times$, $15\times$ and $25\times$. CAECseq achieves the lowest average MEC scores and the highest average CPR in all 3 coverage settings. Since the average MEC scores and CPR achieved by CAECseq and GAEseq significantly outperform the same metrics achieved by other competing algorithms, we proceed by comparing these two methods in terms of their runtimes. Table 2 reports CAECseq and GAEseq runtimes (in seconds) for varied sequencing coverage. As shown there, CAECseq is on average about $3\times$ to $7\times$ faster than GAEseq for coverages varying from $5\times$ to $25\times$, also exhibiting much smaller standard deviation of runtimes for all coverage settings; this is

expected since CAECseq allows mini-batch stochastic gradient descent while GAEseq requires full gradient computation. Performance and runtime comparisons for additional sequencing settings can be found in Supplementary Document B.

The performance comparison on simulated biallelic diploid ($k = 2$) data in terms of the MEC score and CPR with details of experiments can be found in Supplementary Document A. The performance comparison on real *Solanum Tuberosum* data in terms of the MEC score (CPR cannot be evaluated here because ground truth is unavailable) can be found in Supplementary Document C.

Table 1: Performance comparison of CAECseq, HapCompass, H-PoP, AltHap and GAEseq on *Solanum Tuberosum* semi-experimental data for sequencing coverage $5\times$, $15\times$ and $25\times$.

| Coverage | | MEC Mean | Std | CPR Mean | Std |
|---|---|---|---|---|---|
| 5 | CAECseq | **45.6** | **9.3** | **0.85** | **0.02** |
| | HapCompass | 655.2 | 154.6 | 0.61 | 0.04 |
| | H-PoP | 54.9 | 15.9 | 0.83 | 0.06 |
| | AltHap | 418.3 | 114.5 | 0.63 | 0.05 |
| | GAEseq | 49.2 | 16.8 | 0.84 | 0.03 |
| 15 | CAECseq | **87.9** | **39.7** | **0.90** | 0.05 |
| | HapCompass | 2040.5 | 730.9 | 0.61 | 0.07 |
| | H-PoP | 177.4 | 52.7 | 0.86 | 0.07 |
| | AltHap | 594.0 | 167.6 | 0.69 | 0.05 |
| | GAEseq | 176.1 | 49.4 | 0.88 | **0.04** |
| 25 | CAECseq | **101.8** | **44.8** | **0.95** | 0.04 |
| | HapCompass | 4074.8 | 904.0 | 0.63 | 0.04 |
| | H-PoP | 318.6 | 123.8 | 0.84 | 0.06 |
| | AltHap | 509.2 | 181.2 | 0.75 | 0.04 |
| | GAEseq | 204.0 | 118.8 | 0.84 | **0.03** |

Table 2: Run time comparison between CAECseq and GAEseq in seconds on *Solanum Tuberosum* semi-experimental data for sequencing coverage $5\times$, $15\times$ and $25\times$.

| Coverage | CAECseq Time (s) Mean | Std | GAEseq Time (s) Mean | Std |
|---|---|---|---|---|
| 5 | **214.8** | **8.0** | 603.8 | 32.6 |
| 15 | **270.6** | **20.9** | 1578.2 | 144.4 |
| 25 | **311.9** | **16.9** | 2278.0 | 254.4 |

## 3.2 Performance comparison on real HIV-1 data

Next, we compare the performance of CAECseq with state-of-the-art viral quasispecies reconstruction methods including GAEseq (Ke and Vikalo, 2020) (a graph auto-encoder); TenSQR (Ahn, Ke, and Vikalo, 2018), a tensor factorization framework which successively removes reads after using them to reconstruct a dominant strain; PredHaplo (Prabhakaran et al., 2014), a method that relies on a non-parametric Bayesian model, and aBayesQR (Ahn and Vikalo, 2017), a sequential Bayesian inference method on real HIV-1 5-virus-mix data. The ground truth for 5 HIV-1 strains can be obtained from `https://bmda.dmi.unibas.ch/software.html`; the ground truth allows us to evaluate CPR in addition to MEC scores. According to (Di Giallonardo et al., 2014), the relative abundance of the 5 HIV-1 strains is between $10\%$ and $30\%$, and the pairwise Hamming distance between 2 strains is approximately between $2.61\%$ and $8.45\%$. Illumina's MiSeq paired-end reads of length $2 \times 250$ bp are aligned to HIV-$1_{HXB2}$ reference genome. The MEC improvement rate used to estimate the number of strains is set to 0.09 following (Ahn and Vikalo, 2017; Ke and Vikalo, 2020). Reads with mapping score lower than 60 and length shorter than 150 bp are filtered out for quality control. Gene-wise reconstruction of HIV-1 strains is then performed by CAECseq and the selected competing methods. Table 3 reports the number of reads, the length of genes and the number of SNPs for 13 HIV-1 genes. Table 4 shows CPR of each HIV-1 strain, as well as PredProp (the ratio of estimated and true numbers of viral strains) achieved by CAECseq and the competing methods

for different HIV-1 genes. Out of 13 HIV-1 genes, CAECseq perfectly reconstructs all the HIV-1 strains in 9 genes while the closest competitor, GAEseq, correctly reconstructs strains in 8 genes. Other competing methods perfectly reconstruct viral strains in 5 or 6 genes. Note that no method can correctly reconstruct the strains in 4 genes (vpu, gp120, gp41 and nef). As noted in (Ke and Vikalo, 2020), this may be due to translocations of short segments in those genes, causing mismatch between the 5 HIV-1 strains reconstructed by (Di Giallonardo et al., 2014) and the actual ground truth.

The performance comparison of CAECseq and the selected methods on simulated 5-strain viral quasispecies data with different sequencing error rates and varying levels of diversity in terms of MEC, CPR, recall, precision, *Predicted Proportion* and *Jensen–Shannon divergence* can be found in Supplementary Document D. Finally, we apply CAECseq to the problem of reconstructing complete strains of Zika virus; details can be found in Supplementary Document E.

Table 3: Number of reads, length of genes and number of SNPs of HIV-1 genes.

|  | p17 | p24 | p2-p6 | PR | RT | RNase | int | vif | vpr | vpu | gp120 | gp41 | nef |
|---|---|---|---|---|---|---|---|---|---|---|---|---|---|
| Number of reads | 40670 | 63873 | 48089 | 60781 | 156261 | 72858 | 83619 | 39987 | 33494 | 33747 | 69534 | 62428 | 23697 |
| Length of genes | 396 | 693 | 413 | 297 | 1320 | 350 | 866 | 578 | 291 | 248 | 1533 | 1037 | 620 |
| Number of SNPs | 47 | 45 | 31 | 18 | 87 | 37 | 46 | 62 | 32 | 35 | 215 | 132 | 89 |

Table 4: Performance comparison of CAECseq, GAEseq, PredictHap, TenSQR and aBayesQR on real HIV-1 data. Genes where all the strains are perfectly reconstructed are marked in bold.

|  |  | p17 | p24 | p2-p6 | PR | RT | RNase | int | vif | vpr | vpu | gp120 | gp41 | nef |
|---|---|---|---|---|---|---|---|---|---|---|---|---|---|---|
| CAECseq | PredProp | **1** | **1** | **1** | **1** | **1.2** | **1** | **1** | **1** | **1** | 1.2 | 1 | 1 | 1 |
|  | $\text{CPR}_{HXB2}$ | **100** | **100** | **100** | **100** | **100** | **100** | **100** | **100** | **100** | 100 | 96.7 | 97.7 | 100 |
|  | $\text{CPR}_{89.6}$ | **100** | **100** | **100** | **100** | **100** | **100** | **100** | **100** | **100** | 98.4 | 99.0 | 100 | 99.0 |
|  | $\text{CPR}_{JR-CSF}$ | **100** | **100** | **100** | **100** | **100** | **100** | **100** | **100** | **100** | 100 | 99.3 | 100 | 99.4 |
|  | $\text{CPR}_{NL4-3}$ | **100** | **100** | **100** | **100** | **100** | **100** | **100** | **100** | **100** | 100 | 97.2 | 100 | 99.8 |
|  | $\text{CPR}_{YU2}$ | **100** | **100** | **100** | **100** | **100** | **100** | **100** | **100** | **100** | 100 | 99.7 | 100 | 98.1 |
| GAEseq | PredProp | **1** | 1 | **1** | **1** | **1.2** | **1** | **1** | **1** | **1** | 1.2 | 1 | 1 | 1 |
|  | $\text{CPR}_{HXB2}$ | **100** | 99.4 | **100** | **100** | **100** | **100** | **100** | **100** | **100** | 100 | 96.2 | 96.7 | 100 |
|  | $\text{CPR}_{89.6}$ | **100** | 99.4 | **100** | **100** | **100** | **100** | **100** | **100** | **100** | 99.2 | 99.4 | 100 | 98.2 |
|  | $\text{CPR}_{JR-CSF}$ | **100** | 100 | **100** | **100** | **100** | **100** | **100** | **100** | **100** | 100 | 99.9 | 100 | 99.3 |
|  | $\text{CPR}_{NL4-3}$ | **100** | 100 | **100** | **100** | **100** | **100** | **100** | **100** | **100** | 100 | 100 | 100 | 99.8 |
|  | $\text{CPR}_{YU2}$ | **100** | 100 | **100** | **100** | **100** | **100** | **100** | **100** | **100** | 100 | 99.6 | 100 | 98.1 |
| TenSQR | PredProp | **1** | 1.6 | **1** | 1 | 1.4 | **1** | **1** | **1** | **1** | 1.6 | 2.2 | 1.2 | 0.8 |
|  | $\text{CPR}_{HXB2}$ | **100** | 98.9 | **100** | 100 | 99.2 | **100** | **100** | **100** | **100** | 92.8 | 96.0 | 99.0 | 0 |
|  | $\text{CPR}_{89.6}$ | **100** | 100 | **100** | 100 | 98.0 | **100** | **100** | **100** | **100** | 94.0 | 97.2 | 100 | 95.7 |
|  | $\text{CPR}_{JR-CSF}$ | **100** | 100 | **100** | 100 | 100 | **100** | **100** | **100** | **100** | 100 | 98.3 | 97.7 | 99.8 |
|  | $\text{CPR}_{NL4-3}$ | **100** | 99.3 | **100** | 100 | 99.5 | **100** | **100** | **100** | **100** | 100 | 99.8 | 99.5 | 99.7 |
|  | $\text{CPR}_{YU2}$ | **100** | 99.3 | **100** | 99.7 | 99.7 | **100** | **100** | **100** | **100** | 100 | 94.9 | 100 | 98.6 |
| PredictHap | PredProp | **1** | 0.6 | **1** | **1** | **1** | 0.8 | 0.8 | 0.8 | **1** | 0.8 | 0.8 | 0.8 | 0.8 |
|  | $\text{CPR}_{HXB2}$ | **100** | 0 | **100** | **100** | **100** | 98.9 | 100 | 100 | **100** | 93.2 | 0 | 0 | 0 |
|  | $\text{CPR}_{89.6}$ | **100** | 100 | **100** | **100** | **100** | 100 | 99.8 | 100 | **100** | 0 | 97.8 | 100 | 98.8 |
|  | $\text{CPR}_{JR-CSF}$ | **100** | 100 | **100** | **100** | **100** | 100 | 100 | 100 | **100** | 100 | 99.7 | 100 | 100 |
|  | $\text{CPR}_{NL4-3}$ | **100** | 99.1 | **100** | **100** | **100** | 100 | 100 | 100 | **100** | 100 | 100 | 100 | 100 |
|  | $\text{CPR}_{YU2}$ | **100** | 0 | **100** | **100** | **100** | 0 | 0 | 0 | **100** | 100 | 98.6 | 100 | 100 |
| aBayesQR | PredProp | 1 | 1 | **1** | **1** | 1 | **1** | 1 | 1 | **1.2** | 1 | 0.8 | 0.8 | 1.2 |
|  | $\text{CPR}_{HXB2}$ | **100** | 99.4 | **100** | **100** | 98.5 | **100** | 99.9 | 100 | **100** | 99.6 | 98 | 0 | 95.8 |
|  | $\text{CPR}_{89.6}$ | **100** | 98.7 | **100** | **100** | 98.6 | **100** | 100 | 100 | **100** | 92 | 96.5 | 98.9 | 95.5 |
|  | $\text{CPR}_{JR-CSF}$ | **100** | 99.6 | **100** | **100** | 99 | **100** | 100 | 100 | **100** | 98.8 | 97.7 | 99.1 | 98.2 |
|  | $\text{CPR}_{NL4-3}$ | **100** | 100 | **100** | **100** | 98.9 | **100** | 100 | 99.8 | **100** | 100 | 96.3 | 98.8 | 100 |
|  | $\text{CPR}_{YU2}$ | **100** | 99.7 | **100** | **100** | 99.2 | **100** | 99.5 | 99.7 | **100** | 100 | 0 | 98.6 | 99.2 |

## 4 Conclusions

We proposed a novel method for haplotype assembly and viral quasispecies reconstruction from high-throughput sequencing data based on a convolutional auto-encoder with a clustering layer. The convolutional auto-encoder is first trained to project reads to a low dimensional feature space; the clustering layer is initialized by implementing k-means on the learned embedded features. The auto-encoder and the clustering layer with a judiciously chosen target distribution of read origins are then trained simultaneously to group together reads originating from the same genomic strain without distorting the learned feature space. Benchmarking on simulated, semi-experimental and real data show that CAECseq generally outperforms state-of-the-art methods in both haplotype assembly and viral quasispecies reconstruction tasks. We attribute the strong performance of CAECseq in part to its ability to preserve and exploit spatial relationships between SNPs. Moreover, by admitting mini-batch stochastic gradient descent, the runtimes of CAECseq are significantly lower than GAEseq, the only other existing neural network based method for the problem.

## Acknowledgement

This work was funded in part by the NSF grants CCF 1618427 and 2027773.

## Broader Impact

Reconstruction of haplotypes and viral quasispecies from sequencing data are challenging due to limitations of high-throughput sequencing platforms and the large dimensions of these problems. In-depth studies of haplotypes are critical for understanding individual's susceptibility to a broad range of chronic and acute diseases. Moreover, studies of viral quasispecies provide insight into viral dynamics and offer guidance in the development of effective medical therapeutics for diseases caused by RNA viruses such as HIV, HCV, Zika, coronavirus and so on. Therefore, the results of work presented in this manuscript have potential to benefit society by aiding medical research. Potential ethical concern may arise should the proposed haplotype reconstruction techniques be adopted to prenatal testing.

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
