[Supplementary Material]

# A Convolutional Auto-Encoder for Haplotype Assembly and Viral Quasispecies Reconstruction

**Ziqi Ke and Haris Vikalo**
Department of Electrical and Computer Engineering
The University of Texas at Austin
ziqike@utexas.edu, hvikalo@ece.utexas.edu

## Supplementary Document A: Performance comparison on simulated biallelic diploid data

We further benchmarked the performance of CAECseq on synthetic biallelic diploid ($k = 2$) data in terms of the MEC score and CPR. The simulated data is created by generating a reference genome, simulating mutations to generate haplotypes, generating reads with shotgun sequencing, aligning reads to the reference genome, and, finally, calling SNPs. Specifically, the reference genome 5000 base pairs (bp) long is generated by selecting one of four nucleotides with uniform distribution for each genomic position. Haplotypes are then synthesized using *Haplogenerator* (Motazedi et al., 2018) which imputes independent mutations on the reference genome according to a log-normal distribution. The mean distance between mutations and the standard deviation are set to 10 and 3, respectively, generating haplotypes with length approximately 250. Illumina's MiSeq reads of length $2 \times 250$ bp with mean inner distance 50 bp and standard deviation 10 bp are generated utilizing *ART* (Huang et al., 2012), where the sequencing error rate is automatically inferred by this tool based on the massive amounts of sequencing data. Read alignment is performed using *BWA-MEM* (Li and Durbin, 2009), where the reads with mapping scores lower than 40 are filtered out for quality control. SNP positions are determined by comparing the frequency of the alternative allele at any given site with a predefined threshold. Read coverage is varied from $5\times$ to $40\times$ in steps of $5\times$, yielding read numbers varying from about $100$ to $800$. For each coverage setting, 10 data samples are generated and used to compute the mean and standard deviation of MEC scores and CPR achieved by CAECseq and the selected competing methods. In particular, performance of CAECseq is compared with state-of-the-art methods including HapCompass (Aguiar and Istrail, 2012), a method based on graph theory; H-PoP (Xie et al., 2016), an algorithm utilizing dynamic programming; AltHap (Hashemi, Zhu, and Vikalo, 2018), an algorithm using matrix factorization; GAEseq (Ke and Vikalo, 2020), a framework based on a graph auto-encoder; and HapCUT2 (Edge, Bafna, and Bansal, 2017), a maximum-likelihood-based tool. Table 1 shows the results of the aforementioned benchmarking tests. CAECseq achieves the lowest mean and standard deviation of MEC scores, and the highest mean CPR in all settings. Among 8 coverage settings, CAECseq achieves the lowest standard deviation of CPR in 5 settings. Note that the MEC score grows with coverage because higher coverage implies more reads.

Table 1: Performance comparison of CAECseq, HapCompass, H-PoP, AltHap, GAEseq and Hap-CUT2 on simulated diploid data.

| Coverage | | MEC Mean | Std | CPR Mean | Std |
|---|---|---|---|---|---|
| 5 | CAECseq | **16.7** | **4.0** | **0.9582** | 0.0995 |
| | HapCompass | 597.4 | 133.1 | 0.7377 | **0.0636** |
| | H-PoP | 53.2 | 19.3 | 0.9391 | 0.0886 |
| | AltHap | 370.7 | 239.0 | 0.6377 | 0.1181 |
| | GAEseq | 17.7 | 4.2 | 0.9517 | 0.0912 |
| | HapCUT2 | 40.7 | 18.2 | 0.9477 | 0.0900 |
| 10 | CAECseq | **18.9** | **5.4** | **0.9986** | **0.0020** |
| | HapCompass | 1406.0 | 176.8 | 0.7466 | 0.0203 |
| | H-PoP | 97.8 | 30.4 | 0.9807 | 0.0076 |
| | AltHap | 84.5 | 91.4 | 0.8793 | 0.1455 |
| | GAEseq | 20.1 | 6.7 | 0.9896 | 0.0060 |
| | HapCUT2 | 70.0 | 20.9 | 0.9882 | 0.0056 |
| 15 | CAECseq | **29.5** | **5.5** | **0.9986** | **0.0020** |
| | HapCompass | 1944.2 | 277.5 | 0.7589 | 0.0315 |
| | H-PoP | 155.6 | 44.4 | 0.9797 | 0.0042 |
| | AltHap | 229.9 | 165.7 | 0.7675 | 0.1856 |
| | GAEseq | 35.4 | 6.3 | 0.9926 | 0.0030 |
| | HapCUT2 | 114.5 | 45.2 | 0.9875 | 0.0036 |
| 20 | CAECseq | **34.4** | **4.4** | **0.9994** | **0.0013** |
| | HapCompass | 2624.8 | 322.6 | 0.7464 | 0.0245 |
| | H-PoP | 162.5 | 42.3 | 0.9851 | 0.0036 |
| | AltHap | 173.0 | 199.9 | 0.9137 | 0.1228 |
| | GAEseq | 40.6 | 5.8 | 0.9958 | 0.0025 |
| | HapCUT2 | 117.1 | 30.6 | 0.9915 | 0.0027 |
| 25 | CAECseq | **40.7** | **5.4** | **0.9704** | 0.0887 |
| | HapCompass | 2798.4 | 766.6 | 0.7415 | 0.1118 |
| | H-PoP | 229.2 | 86.5 | 0.9540 | **0.0877** |
| | AltHap | 195.7 | 278.7 | 0.9045 | 0.1535 |
| | GAEseq | 58.7 | 6.2 | 0.9303 | 0.1414 |
| | HapCUT2 | 163.1 | 56.8 | 0.9518 | 0.1182 |
| 30 | CAECseq | **55.6** | **7.8** | **0.9994** | **0.0013** |
| | HapCompass | 5529.8 | 4207.1 | 0.6727 | 0.2270 |
| | H-PoP | 367.3 | 76.7 | 0.9775 | 0.0061 |
| | AltHap | 419.6 | 397.7 | 0.8349 | 0.1760 |
| | GAEseq | 75.6 | 9.2 | 0.9896 | 0.0026 |
| | HapCUT2 | 266.2 | 49.4 | 0.9846 | 0.0041 |
| 35 | CAECseq | **66.4** | **11.1** | **0.9996** | 0.0007 |
| | HapCompass | 4966.7 | 754.2 | 0.7436 | 0.0329 |
| | H-PoP | 383.3 | 104.2 | 0.9798 | 0.0068 |
| | AltHap | 294.2 | 345.8 | 0.9306 | 0.1074 |
| | GAEseq | 69.2 | 12.4 | 0.9994 | **0.0006** |
| | HapCUT2 | 268.9 | 59.7 | 0.9882 | 0.0031 |
| 40 | CAECseq | **69.2** | **8.2** | **0.9995** | **0.0012** |
| | HapCompass | 7462.4 | 5499.5 | 0.6646 | 0.2243 |
| | H-PoP | 394.1 | 69.2 | 0.9818 | 0.0022 |
| | AltHap | 436.8 | 589.2 | 0.9355 | 0.1033 |
| | GAEseq | 78.6 | 10.8 | 0.9991 | 0.0018 |
| | HapCUT2 | 268.5 | 85.7 | 0.9889 | 0.0039 |

## Supplementary Document B: Performance comparison on semi-experimental *Solanum Tuberosum* data

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

**Supplementary Document C: Performance comparison on real *Solanum Tuberosum* data**

The performance of CAECseq is further tested on the real *Solanum Tuberosum* chromosome 5 data (NCBI accession SRR6173308). Ten genomic regions are randomly selected as the reference genome to generate 10 data samples. Illumina HiSeq 2000 paired-end reads with quality score higher than 40 are then aligned to the selected genomic regions using *BWA-MEM* (Li and Durbin, 2009), followed by the SNP calling step. Table 4 shows the number of reads, the length of genes and the number of SNPs in 10 *Solanum Tuberosum* regions. Since for real data the ground truth is unavailable, we only evaluate MEC scores and show them in Table 5. As seen there, CAECseq achieves the lowest MEC in 7 out of 10 regions.

Table 4: Number of reads, length of genes and number of SNPs of 10 real *Solanum Tuberosum* regions.

| Region | 1 | 2 | 3 | 4 | 5 | 6 | 7 | 8 | 9 | 10 |
|---|---|---|---|---|---|---|---|---|---|---|
| Number of reads | 240 | 389 | 274 | 115 | 141 | 398 | 295 | 284 | 489 | 449 |
| Length of genes | 5035 | 5032 | 5908 | 5981 | 5757 | 5877 | 5603 | 5608 | 5640 | 7573 |
| Number of SNVs | 294 | 238 | 83 | 23 | 176 | 198 | 456 | 424 | 236 | 410 |

Table 5: Performance comparison of CAECseq, HapCompass, H-PoP, AltHap and GAEseq on Real *Solanum Tuberosum* data in terms of MEC.

| Region | CAECseq | HapCompass | H-PoP | AltHap | GAEseq |
|---|---|---|---|---|---|
| 1 | **229** | 1001 | 235 | 516 | 231 |
| 2 | **393** | 1105 | 460 | 557 | 406 |
| 3 | 103 | 1098 | 140 | 241 | **97** |
| 4 | **1** | 28 | 4 | 11 | 2 |
| 5 | 172 | 1084 | **168** | 342 | 180 |
| 6 | **859** | 6372 | 917 | 1124 | 873 |
| 7 | **522** | 5298 | 571 | 986 | 558 |
| 8 | **430** | 5246 | 613 | 1238 | 441 |
| 9 | 593 | 2250 | **534** | 947 | 592 |
| 10 | **698** | 2578 | 751 | 1059 | 712 |
| Mean | **400.0** | 2606.0 | 441.1 | 702.1 | 409.2 |
| Std | **260.9** | 2111.7 | 277.4 | 401.1 | 266.6 |

## Supplementary Document D: Performance comparison on simulated viral quasispecies data

The performance of CAECseq is further tested in an application to the reconstruction of viral quasispecies on a dataset with 5 synthetic strains. In addition to the MEC score and CPR, performance of methods for viral quasispecies reconstruction is expressed in terms of *recall*, the proportion of reconstructed viral strains that match the true viral strains; *precision*, the proportion of strains that are perfectly reconstructed in the reconstructed strains; *Predicted Proportion* (PredProp), the ratio of estimated and true numbers of viral strains, and *Jensen–Shannon divergence* (JSD), which measures the difference between the estimated frequencies of strains and the true frequencies, i.e.

$$\text{JSD}(P||Q) = \frac{1}{2}D(P||M) + \frac{1}{2}D(Q||M), \tag{1}$$

where $D(\cdot||\cdot)$ denotes Kullback-Leibler (KL) divergence defined as $D(P||Q) = \sum_i P(i)\log\frac{P(i)}{Q(i)}$, and $M = \frac{1}{2}(P + Q)$ (Ahn, Ke, and Vikalo, 2018). Note that apart from MEC scores, all the performance metrics can be evaluated only when the ground truth is available.

Following (Ahn, Ke, and Vikalo, 2018), the reference genome of length 1300 bp, which is the length of HIV-1 *pol* region, is generated by selecting on each site one of four nucleotides from uniform distribution. Independent mutations on the reference genome are then generated from uniform distribution to synthesize 5 viral strains. Illumina's MiSeq paired-end reads of length $2 \times 250$ bp with mean inner distance 150 bp and standard deviation 30 bp are generated next. *BWA-MEM* (Li and Durbin, 2009) is used for read alignment and reads with mapping scores lower than 40 are filtered out for quality control. Two typical MiSeq sequencing error rates, 0.002 and 0.007, are used to simulate errors and 10 samples with varying diversity (defined as the average pairwise Hamming distance between 2 strains in a viral population) from $1\%$ to $10\%$ with step size $1\%$ are generated independently 10 times for each error rate. The relative abundances of 5 strains are set to 0.5, 0.3, 0.15, 0.04 and 0.01 (setting up the scenario wherein the ability of CAECseq to reconstruct imbalanced viral populations can be tested); the sequencing coverage is set to 500. The number of reads in each sample is 6500 and the number of SNPs varies from approximately 30 bp to 300 bp. Performance of CAECseq is compared with state-of-the-art methods including GAEseq (Ke and Vikalo, 2020) (a graph auto-encoder); TenSQR (Ahn, Ke, and Vikalo, 2018), a tensor factorization framework which successively removes reads after using them to reconstruct a dominant strain; PredHaplo (Prabhakaran et al., 2014), a method that relies on a Dirichlet process mixture model, and aBayesQR (Ahn and Vikalo, 2017), a sequential Bayesian inference method. Performance is measured by means of MEC scores, CPR, recall, precision, PredProp and JSD as defined in Section 2.1; the mean and standard deviation of each performance metric are evaluated by averaging over 10 samples, each with fixed sequencing error rate and diversity. Table 6 and 7 compare the performance of CAECseq and the selected competing methods in terms of MEC scores and CPR, where the sequencing error rate is $\epsilon = 0.002$ and $\epsilon = 0.007$, respectively. PredictHaplo fails to run on some of the samples with low diversity ($1\%$ - $3\%$), and hence only the results where PredictHaplo succeeds in running on all 10 samples are shown. CAECseq outperforms all the other selected methods at almost all levels of diversity in terms of the mean and standard deviation of MEC scores and CPR. CPR achieved by CAECseq is typically very close to 1, validating its ability to accurately reconstruct viral strains even at low diversities. Table 8 and 9 compare the performance of CAECseq and the competing methods in terms of recall and precision, where the sequencing error rate is $\epsilon = 0.002$ and $\epsilon = 0.007$, respectively. For sequencing error rates $\epsilon = 0.002$ and $\epsilon = 0.007$, CAECseq outperforms all the other selected methods for 9 and 10 out of 10 levels of diversity, respectively. CAECseq also outperforms all the competing methods at diversities $1\%$ - $3\%$, with PredHaplo achieving high precision rate at diversity $\geq 5\%$ only because PredHaplo underestimates the viral population size and often fails to reconstruct viral strains whose relative abundance is lower than $15\%$. Table 10 and 11 compare the performance of CAECseq and competing methods in terms of PredProp and JSD for sequencing error rate $\epsilon = 0.002$ and $\epsilon = 0.007$, respectively. On the task of estimating the viral population size, CAECseq performs the best even at low diversities (those in $1\%$ - $2\%$ range), reflecting the ability of CAECseq to distinguish highly similar strains by capturing local features of the sequencing reads. At diversity $\geq 3\%$, CAECseq, GAEseq, TenSQR perform similarly – correctly estimating the population size and finding the correct origin of reads – while aBayesQR and PredictHaplo tend to underestimate the population size.

Table 6: Performance comparison of CAECseq, GAEseq, TenSQR, PredHaplo and aBayesQR on simulated 5-virus-mix data with sequencing error $\epsilon = 2 \times 10^{-3}$ in terms of MEC and CPR.

| Diversity | | MEC Mean | Std | CPR Mean | Std |
|---|---|---|---|---|---|
| 1 | CAECseq | **43.6** | 8.9 | **0.9997** | 0.0006 |
| | GAEseq | 44.5 | **8.8** | 0.9996 | 0.0007 |
| | TenSQR | 45.4 | 8.9 | 0.9995 | 0.0006 |
| | PredHaplo | - | - | - | - |
| | aBayesQR | 424.8 | 768.8 | 0.9394 | 0.0913 |
| 2 | CAECseq | **95.4** | **8.4** | 0.9998 | 0.0003 |
| | GAEseq | 105.2 | 9.5 | 0.9998 | 0.0002 |
| | TenSQR | 98.9 | 8.7 | 0.9996 | 0.0002 |
| | PredHaplo | - | - | - | - |
| | aBayesQR | 674.4 | 1253.5 | 0.9395 | 0.0918 |
| 3 | CAECseq | **139.3** | **9.8** | **0.9997** | **0.0002** |
| | GAEseq | 187.2 | 11.1 | 0.9993 | 0.0004 |
| | TenSQR | 152.8 | 10.9 | 0.9995 | 0.0003 |
| | PredHaplo | - | - | - | - |
| | aBayesQR | 461.5 | 245.4 | 0.9395 | 0.0914 |
| 4 | CAECseq | 224.9 | 24.5 | 0.9996 | **0.0002** |
| | GAEseq | 227.0 | **20.6** | 0.9994 | 0.0005 |
| | TenSQR | **205.3** | 22.7 | 0.9996 | 0.0004 |
| | PredHaplo | 4241.2 | 2458.2 | 0.8127 | 0.3847 |
| | aBayesQR | 3890.4 | 6702.2 | 0.8789 | 0.0983 |
| 5 | CAECseq | **247.5** | **13.7** | **0.9996** | **0.0001** |
| | GAEseq | 251.4 | 15.9 | 0.9995 | 0.0003 |
| | TenSQR | 253.9 | 16.4 | 0.9995 | 0.0003 |
| | PredHaplo | 543.1 | 321.7 | 0.9902 | 0.0028 |
| | aBayesQR | 980.0 | 627.9 | 0.9579 | 0.0790 |
| 6 | CAECseq | **254.6** | **11.4** | **0.9998** | **0.0002** |
| | GAEseq | 357.6 | 15.2 | 0.9995 | 0.0003 |
| | TenSQR | 303.4 | 15.8 | 0.9995 | 0.0004 |
| | PredHaplo | 647.2 | 214.1 | 0.9890 | 0.0009 |
| | aBayesQR | 5215.3 | 7048.8 | 0.8580 | 0.1556 |
| 7 | CAECseq | **368.7** | **28.4** | **0.9997** | 0.0002 |
| | GAEseq | 383.5 | 32.1 | 0.9996 | 0.0003 |
| | TenSQR | 373.5 | 31.1 | 0.9995 | 0.0002 |
| | PredHaplo | 847.8 | 743.3 | 0.9871 | 0.0009 |
| | aBayesQR | 2200.5 | 2387.0 | 0.9568 | 0.0795 |
| 8 | CAECseq | 394.4 | 23.8 | 0.9994 | 0.0003 |
| | GAEseq | **387.5** | **22.5** | **0.9995** | 0.0004 |
| | TenSQR | 396.3 | 24.4 | 0.9994 | 0.0003 |
| | PredHaplo | 1242.2 | 1342.1 | 0.9851 | 0.0010 |
| | aBayesQR | 3690.1 | 4536.0 | 0.9165 | 0.0988 |
| 9 | CAECseq | **447.3** | **16.6** | **0.9992** | **0.0005** |
| | GAEseq | 468.8 | 18.5 | 0.9991 | 0.0007 |
| | TenSQR | 456.8 | 20.9 | 0.9990 | 0.0008 |
| | PredHaplo | 1679.0 | 1104.2 | 0.9827 | 0.0012 |
| | aBayesQR | 4182.6 | 4413.2 | 0.8381 | 0.1479 |
| 10 | CAECseq | 510.3 | 35.9 | **0.9989** | 0.0007 |
| | GAEseq | 678.0 | **33.7** | 0.9984 | 0.0006 |
| | TenSQR | **503.0** | 35.5 | 0.9988 | 0.0006 |
| | PredHaplo | 2248.1 | 1798.5 | 0.9793 | 0.0041 |
| | aBayesQR | 3740.6 | 2553.4 | 0.8774 | 0.0968 |

Table 7: Performance comparison of CAECseq, GAEseq, TenSQR, PredHaplo and aBayesQR on simulated 5-virus-mix data with sequencing error $\epsilon = 7 \times 10^{-3}$ in terms of MEC and CPR.

| Diversity | | MEC Mean | Std | CPR Mean | Std |
|---|---|---|---|---|---|
| 1 | CAECseq | **200.9** | **31.8** | **0.9992** | **0.0007** |
| | GAEseq | 280.9 | 35.5 | 0.9980 | 0.0009 |
| | TenSQR | 210.8 | 32.9 | 0.9990 | 0.0009 |
| | PredHaplo | - | - | - | - |
| | aBayesQR | 798.2 | 445.0 | 0.9582 | 0.0795 |
| 2 | CAECseq | **397.4** | 46.7 | **0.9993** | **0.0003** |
| | GAEseq | 513.1 | 50.7 | 0.9991 | 0.0004 |
| | TenSQR | 419.6 | **45.8** | 0.9992 | 0.0004 |
| | PredHaplo | - | - | - | - |
| | aBayesQR | 2814.8 | 1605.8 | 0.8183 | 0.1391 |
| 3 | CAECseq | **548.2** | **25.6** | 0.9993 | 0.0005 |
| | GAEseq | 553.0 | 27.9 | 0.9993 | 0.0006 |
| | TenSQR | 575.0 | 29.9 | 0.9992 | 0.0005 |
| | PredHaplo | - | - | - | - |
| | aBayesQR | 5836.4 | 5585.7 | 0.8567 | 0.1784 |
| 4 | CAECseq | 694.8 | **47.8** | **0.9997** | **0.0002** |
| | GAEseq | **682.5** | 49.1 | 0.9995 | 0.0005 |
| | TenSQR | 750.4 | 48.2 | 0.9994 | 0.0004 |
| | PredHaplo | 2886.0 | 2204.6 | 0.8722 | 0.3254 |
| | aBayesQR | 2903.3 | 1919.6 | 0.8573 | 0.1266 |
| 5 | CAECseq | 978.4 | 56.1 | 0.9989 | 0.0005 |
| | GAEseq | 1087.6 | **55.7** | 0.9989 | 0.0006 |
| | TenSQR | **941.4** | 58.7 | **0.9990** | 0.0005 |
| | PredHaplo | 3980.1 | 1247.6 | 0.9904 | 0.0017 |
| | aBayesQR | 6920.5 | 11069.4 | 0.8742 | 0.0957 |
| 6 | CAECseq | **1028.7** | 52.8 | **0.9990** | **0.0006** |
| | GAEseq | 1039.5 | 53.5 | 0.9988 | 0.0007 |
| | TenSQR | 1132.2 | **51.1** | 0.9987 | 0.0007 |
| | PredHaplo | 4578.4 | 2217.2 | 0.9885 | 0.0017 |
| | aBayesQR | 6026.0 | 3510.0 | 0.7771 | 0.1636 |
| 7 | CAECseq | **1154.7** | **65.1** | **0.9992** | **0.0005** |
| | GAEseq | 1280.2 | 75.5 | 0.9990 | 0.0007 |
| | TenSQR | 1308.5 | 70.0 | 0.9988 | 0.0007 |
| | PredHaplo | 5421.2 | 2179.3 | 0.9870 | 0.0008 |
| | aBayesQR | 11235.5 | 3388.6 | 0.7356 | 0.1245 |
| 8 | CAECseq | **1351.4** | **67.2** | **0.9992** | 0.0004 |
| | GAEseq | 1394.1 | 68.3 | 0.9991 | 0.0004 |
| | TenSQR | 1482.6 | 67.3 | 0.9989 | 0.0005 |
| | PredHaplo | 5147.2 | 1987.4 | 0.9849 | 0.0011 |
| | aBayesQR | 10349.5 | 8523.0 | 0.7567 | 0.1462 |
| 9 | CAECseq | 1543.4 | 86.2 | 0.9991 | 0.0009 |
| | GAEseq | **1538.0** | 85.8 | **0.9992** | 0.0008 |
| | TenSQR | 1641.0 | **79.3** | 0.9986 | 0.0008 |
| | PredHaplo | 4718.3 | 1479.5 | 0.9828 | 0.0011 |
| | aBayesQR | 11599.9 | 15032.4 | 0.7750 | 0.1052 |
| 10 | CAECseq | **1246.9** | **51.5** | **0.9988** | **0.0007** |
| | GAEseq | 1877.5 | 66.2 | 0.9978 | 0.0010 |
| | TenSQR | 1796.4 | 68.3 | 0.9980 | 0.0009 |
| | PredHaplo | 6477.8 | 2976.4 | 0.9795 | 0.0035 |
| | aBayesQR | 7332.3 | 4275.5 | 0.7945 | 0.1209 |

Table 8: Performance comparison of CAECseq, GAEseq, TenSQR, PredHaplo and aBayesQR on simulated 5-virus-mix data with sequencing error $\epsilon = 2 \times 10^{-3}$ in terms of recall and precision.

| Diversity | | Recall Mean | Std | Precision Mean | Std |
|---|---|---|---|---|---|
| 1 | CAECseq | **0.85** | 0.18 | 0.76 | 0.19 |
| | GAEseq | 0.78 | 0.24 | 0.55 | 0.22 |
| | TenSQR | 0.80 | 0.22 | 0.55 | 0.23 |
| | PredHaplo | - | - | - | - |
| | aBayesQR | 0.80 | **0.09** | **0.86** | **0.13** |
| 2 | CAECseq | **0.84** | **0.16** | **0.80** | **0.14** |
| | GAEseq | 0.80 | 0.18 | 0.70 | 0.19 |
| | TenSQR | 0.82 | 0.19 | 0.71 | 0.19 |
| | PredHaplo | - | - | - | - |
| | aBayesQR | 0.78 | 0.17 | 0.79 | 0.15 |
| 3 | CAECseq | **0.80** | 0.12 | **0.80** | 0.12 |
| | GAEseq | 0.72 | 0.10 | 0.72 | 0.10 |
| | TenSQR | 0.72 | 0.10 | 0.72 | 0.10 |
| | PredHaplo | - | - | - | - |
| | aBayesQR | 0.74 | **0.09** | 0.79 | 0.15 |
| 4 | CAECseq | 0.84 | **0.10** | 0.84 | **0.10** |
| | GAEseq | 0.84 | 0.11 | 0.84 | 0.11 |
| | TenSQR | 0.82 | 0.11 | 0.82 | 0.11 |
| | PredHaplo | 0.56 | 0.29 | 0.74 | 0.37 |
| | aBayesQR | 0.64 | 0.22 | 0.68 | 0.15 |
| 5 | CAECseq | **0.80** | **0.11** | 0.80 | **0.11** |
| | GAEseq | 0.72 | 0.15 | 0.72 | 0.15 |
| | TenSQR | 0.74 | 0.16 | 0.74 | 0.16 |
| | PredHaplo | 0.71 | 0.12 | **0.91** | 0.15 |
| | aBayesQR | 0.70 | 0.16 | 0.71 | 0.19 |
| 6 | CAECseq | **0.78** | **0.10** | 0.78 | **0.10** |
| | GAEseq | 0.76 | 0.12 | 0.76 | 0.12 |
| | TenSQR | 0.74 | 0.13 | 0.74 | 0.13 |
| | PredHaplo | 0.72 | 0.13 | **0.90** | 0.17 |
| | aBayesQR | 0.48 | 0.18 | 0.56 | 0.24 |
| 7 | CAECseq | **0.79** | 0.11 | 0.79 | **0.11** |
| | GAEseq | 0.74 | 0.12 | 0.74 | 0.12 |
| | TenSQR | 0.76 | 0.15 | 0.76 | 0.15 |
| | PredHaplo | 0.66 | 0.16 | **0.83** | 0.20 |
| | aBayesQR | 0.62 | 0.11 | 0.63 | 0.17 |
| 8 | CAECseq | **0.76** | **0.12** | 0.76 | **0.12** |
| | GAEseq | 0.72 | 0.14 | 0.72 | 0.14 |
| | TenSQR | 0.70 | 0.13 | 0.70 | 0.13 |
| | PredHaplo | 0.62 | 0.20 | **0.78** | 0.25 |
| | aBayesQR | 0.50 | 0.20 | 0.55 | 0.22 |
| 9 | CAECseq | **0.64** | 0.12 | 0.64 | 0.12 |
| | GAEseq | 0.56 | 0.12 | 0.56 | 0.12 |
| | TenSQR | 0.58 | 0.14 | 0.58 | 0.14 |
| | PredHaplo | 0.59 | 0.20 | **0.74** | 0.25 |
| | aBayesQR | 0.52 | 0.13 | 0.62 | 0.18 |
| 10 | CAECseq | **0.60** | 0.11 | 0.60 | 0.11 |
| | GAEseq | 0.58 | 0.12 | 0.58 | 0.12 |
| | TenSQR | 0.58 | 0.11 | 0.58 | 0.11 |
| | PredHaplo | 0.52 | 0.18 | **0.65** | 0.22 |
| | aBayesQR | 0.42 | 0.14 | 0.48 | 0.18 |

Table 9: Performance comparison of CAECseq, GAEseq, TenSQR, PredHaplo and aBayesQR on simulated 5-virus-mix data with sequencing error $\epsilon = 7 \times 10^{-3}$ in terms of recall and precision.

| Diversity | | Recall Mean | Std | Precision Mean | Std |
|---|---|---|---|---|---|
| 1 | CAECseq | **0.80** | **0.16** | **0.76** | **0.16** |
| | GAEseq | 0.68 | 0.18 | 0.55 | 0.22 |
| | TenSQR | 0.70 | 0.18 | 0.56 | 0.21 |
| | PredHaplo | - | - | - | - |
| | aBayesQR | 0.42 | 0.21 | 0.42 | 0.22 |
| 2 | CAECseq | **0.82** | **0.12** | **0.82** | **0.12** |
| | GAEseq | 0.72 | 0.18 | 0.66 | 0.13 |
| | TenSQR | 0.68 | 0.13 | 0.67 | 0.14 |
| | PredHaplo | - | - | - | - |
| | aBayesQR | 0.42 | 0.17 | 0.49 | 0.25 |
| 3 | CAECseq | **0.78** | **0.10** | **0.78** | **0.10** |
| | GAEseq | 0.72 | 0.11 | 0.76 | 0.13 |
| | TenSQR | 0.76 | 0.12 | 0.76 | 0.12 |
| | PredHaplo | - | - | - | - |
| | aBayesQR | 0.48 | 0.18 | 0.57 | 0.22 |
| 4 | CAECseq | **0.74** | 0.13 | 0.74 | **0.13** |
| | GAEseq | 0.68 | 0.15 | 0.68 | 0.14 |
| | TenSQR | 0.66 | 0.16 | 0.66 | 0.16 |
| | PredHaplo | 0.61 | 0.26 | **0.80** | 0.33 |
| | aBayesQR | 0.50 | 0.13 | 0.58 | 0.16 |
| 5 | CAECseq | **0.72** | **0.11** | 0.72 | **0.11** |
| | GAEseq | 0.66 | 0.13 | 0.62 | 0.13 |
| | TenSQR | 0.64 | 0.12 | 0.64 | 0.12 |
| | PredHaplo | 0.70 | 0.12 | **0.88** | 0.15 |
| | aBayesQR | 0.40 | 0.22 | 0.47 | 0.25 |
| 6 | CAECseq | **0.74** | **0.10** | 0.74 | **0.10** |
| | GAEseq | 0.62 | 0.12 | 0.66 | 0.12 |
| | TenSQR | 0.64 | 0.12 | 0.64 | 0.12 |
| | PredHaplo | 0.70 | 0.12 | **0.89** | 0.15 |
| | aBayesQR | 0.40 | 0.13 | 0.50 | 0.13 |
| 7 | CAECseq | **0.70** | **0.12** | 0.70 | **0.12** |
| | GAEseq | 0.60 | 0.15 | 0.60 | 0.14 |
| | TenSQR | 0.60 | 0.13 | 0.60 | 0.13 |
| | PredHaplo | 0.64 | 0.18 | **0.80** | 0.23 |
| | aBayesQR | 0.40 | 0.13 | 0.58 | 0.25 |
| 8 | CAECseq | **0.70** | 0.10 | 0.70 | **0.10** |
| | GAEseq | 0.66 | 0.10 | 0.63 | 0.13 |
| | TenSQR | 0.64 | 0.12 | 0.64 | 0.12 |
| | PredHaplo | 0.57 | 0.20 | **0.71** | 0.25 |
| | aBayesQR | 0.42 | 0.19 | 0.57 | 0.26 |
| 9 | CAECseq | **0.68** | 0.11 | 0.68 | 0.11 |
| | GAEseq | 0.60 | 0.12 | 0.62 | 0.15 |
| | TenSQR | 0.62 | 0.14 | 0.62 | 0.14 |
| | PredHaplo | 0.58 | 0.19 | **0.73** | 0.23 |
| | aBayesQR | 0.50 | **0.10** | 0.64 | **0.10** |
| 10 | CAECseq | **0.62** | 0.16 | **0.62** | 0.16 |
| | GAEseq | 0.54 | 0.21 | 0.50 | 0.22 |
| | TenSQR | 0.52 | 0.20 | 0.52 | 0.20 |
| | PredHaplo | 0.17 | 0.58 | 0.22 | 0.79 |
| | aBayesQR | 0.38 | **0.14** | 0.49 | 0.23 |

Table 10: Performance comparison of CAECseq, GAEseq, TenSQR, PredHaplo and aBayesQR on simulated 5-virus-mix data with sequencing error $\epsilon = 2 \times 10^{-3}$ in terms of PredProp and JSD.

| Diversity | | PredProp Mean | Std | JSD Mean | Std |
|---|---|---|---|---|---|
| 1 | CAECseq | **1.36** | **0.24** | 0.001 | 0.002 |
| | GAEseq | 1.45 | 0.26 | 0.001 | 0.003 |
| | TenSQR | 1.58 | 0.34 | 0.001 | 0.003 |
| | PredHaplo | - | - | - | - |
| | aBayesQR | 0.94 | 0.09 | 0.001 | 0.002 |
| 2 | CAECseq | **1.07** | **0.02** | 0 | 0 |
| | GAEseq | 1.12 | 0.04 | 0 | 0 |
| | TenSQR | 1.18 | 0.23 | 0 | 0 |
| | PredHaplo | - | - | - | - |
| | aBayesQR | 1.00 | 0.18 | 0.003 | 0.005 |
| 3 | CAECseq | 1 | 0 | 0 | 0 |
| | GAEseq | 1 | 0 | 0 | 0 |
| | TenSQR | 1 | 0 | 0 | 0 |
| | PredHaplo | - | - | - | - |
| | aBayesQR | 0.96 | 0.12 | 0.001 | 0.002 |
| 4 | CAECseq | 1 | 0 | 0 | 0 |
| | GAEseq | 1 | 0 | 0 | 0 |
| | TenSQR | 1 | 0 | 0 | 0 |
| | PredHaplo | 0.62 | 0.31 | 0.083 | 0.064 |
| | aBayesQR | 0.94 | 0.24 | 0.002 | 0.002 |
| 5 | CAECseq | 1 | 0 | 0 | 0 |
| | GAEseq | 1 | 0 | 0 | 0 |
| | TenSQR | 1 | 0 | 0 | 0 |
| | PredHaplo | 0.78 | 0.05 | 0.101 | 0.050 |
| | aBayesQR | 1.00 | 0.13 | 0.001 | 0.001 |
| 6 | CAECseq | 1 | 0 | 0 | 0 |
| | GAEseq | 1 | 0 | 0 | 0 |
| | TenSQR | 1 | 0 | 0 | 0 |
| | PredHaplo | 0.80 | 0 | 0.112 | 0.056 |
| | aBayesQR | 0.88 | 0.18 | 0.005 | 0.007 |
| 7 | CAECseq | 1 | 0 | 0 | 0 |
| | GAEseq | 1 | 0 | 0 | 0 |
| | TenSQR | 1 | 0 | 0 | 0 |
| | PredHaplo | 0.80 | 0 | 0.131 | 0.063 |
| | aBayesQR | 1.02 | 0.17 | 0.001 | 0.001 |
| 8 | CAECseq | 1 | 0 | 0 | 0 |
| | GAEseq | 1 | 0 | 0 | 0 |
| | TenSQR | 1 | 0 | 0 | 0 |
| | PredHaplo | 0.80 | 0 | 0.108 | 0.055 |
| | aBayesQR | 0.92 | 0.10 | 0.003 | 0.006 |
| 9 | CAECseq | 1 | 0 | 0 | 0 |
| | GAEseq | 1 | 0 | 0 | 0 |
| | TenSQR | 1 | 0 | 0 | 0 |
| | PredHaplo | 0.80 | 0 | 0.116 | 0.053 |
| | aBayesQR | 0.86 | 0.18 | 0.005 | 0.007 |
| 10 | CAECseq | 1 | 0 | 0 | 0 |
| | GAEseq | 1 | 0 | 0 | 0 |
| | TenSQR | 1 | 0 | 0 | 0 |
| | PredHaplo | 0.79 | 0.04 | 0.097 | 0.066 |
| | aBayesQR | 0.92 | 0.16 | 0.013 | 0.022 |

Table 11: Performance comparison of CAECseq, GAEseq, TenSQR, PredHaplo and aBayesQR on simulated 5-virus-mix data with sequencing error $\epsilon = 7 \times 10^{-3}$ in terms of PredProp and JSD.

| Diversity | | PredProp Mean | Std | JSD Mean | Std |
|---|---|---|---|---|---|
| 1 | CAECseq | **1.28** | **0.18** | 0.001 | **0.002** |
| | GAEseq | 1.43 | 0.26 | 0.001 | 0.003 |
| | TenSQR | 1.32 | 0.24 | 0.001 | 0.003 |
| | PredHaplo | - | - | - | - |
| | aBayesQR | 1.04 | 0.17 | 0.019 | 0.033 |
| 2 | CAECseq | **1** | **0** | 0 | 0 |
| | GAEseq | 1.10 | 0.05 | 0 | 0 |
| | TenSQR | 1.02 | 0.06 | 0 | 0 |
| | PredHaplo | - | - | - | - |
| | aBayesQR | 0.92 | 0.31 | 0.020 | 0.034 |
| 3 | CAECseq | 1 | 0 | 0 | 0 |
| | GAEseq | 1 | 0 | 0 | 0 |
| | TenSQR | 1 | 0 | 0 | 0 |
| | PredHaplo | - | - | - | - |
| | aBayesQR | 0.86 | 0.18 | 0.017 | 0.027 |
| 4 | CAECseq | 1 | 0 | 0 | 0 |
| | GAEseq | 1 | 0 | 0 | 0 |
| | TenSQR | 1 | 0 | 0 | 0 |
| | PredHaplo | 0.67 | 0.26 | 0.088 | 0.068 |
| | aBayesQR | 0.88 | 0.16 | 0.006 | 0.007 |
| 5 | CAECseq | 1 | 0 | 0 | 0 |
| | GAEseq | 1 | 0 | 0 | 0 |
| | TenSQR | 1 | 0 | 0 | 0 |
| | PredHaplo | 0.79 | 0.04 | 0.101 | 0.060 |
| | aBayesQR | 0.88 | 0.10 | 0.023 | 0.043 |
| 6 | CAECseq | 1 | 0 | 0 | 0 |
| | GAEseq | 1 | 0 | 0 | 0 |
| | TenSQR | 1 | 0 | 0 | 0 |
| | PredHaplo | 0.80 | 0.03 | 0.111 | 0.064 |
| | aBayesQR | 0.80 | 0.20 | 0.008 | 0.008 |
| 7 | CAECseq | 1 | 0 | 0 | 0 |
| | GAEseq | 1 | 0 | 0 | 0 |
| | TenSQR | 1 | 0 | 0 | 0 |
| | PredHaplo | 0.80 | 0 | 0.112 | 0.056 |
| | aBayesQR | 0.74 | 0.13 | 0.009 | 0.007 |
| 8 | CAECseq | 1 | 0 | 0 | 0 |
| | GAEseq | 1 | 0 | 0 | 0 |
| | TenSQR | 1 | 0 | 0 | 0 |
| | PredHaplo | 0.80 | 0 | 0.122 | 0.060 |
| | aBayesQR | 0.76 | 0.15 | 0.010 | 0.007 |
| 9 | CAECseq | 1 | 0 | 0 | 0 |
| | GAEseq | 1 | 0 | 0 | 0 |
| | TenSQR | 1 | 0 | 0 | 0 |
| | PredHaplo | 0.80 | 0 | 0.112 | 0.062 |
| | aBayesQR | 0.78 | 0.11 | 0.006 | 0.006 |
| 10 | CAECseq | 1 | 0 | 0 | 0 |
| | GAEseq | 1 | 0 | 0 | 0 |
| | TenSQR | 1 | 0 | 0 | 0 |
| | PredHaplo | 0.79 | 0.04 | 0.112 | 0.059 |
| | aBayesQR | 0.84 | 0.22 | 0.007 | 0.006 |

**Supplementary Document E: Application to real Zika virus data**

Finally, we apply CAECseq to the problem of reconstructing the full strains of Zika virus using data sampled from an animal 393422 on the fourth day of infection (NCBI accession SRR3332513). Illumina's MiSeq paired-end reads of length $2 \times 300$ bp are aligned to the reference genome (GenBank accession KU681081.3) of length 10807 bp using *BWA-MEM* (Li and Durbin, 2009). Reads with mapping quality score lower than 40 and length shorter than 100 bp are filtered out for quality control, resulting in 591001 reads. Following (Ahn, Ke, and Vikalo, 2018), the full genome is fragmented into regions of length 2500 bp, with consecutive regions overlapped by 501 bp, to enable computationally feasible yet reliable reconstruction of full strains. CAECseq and the same competing methods as in Section 3.4 are implemented in each region to reconstruct sub-strains. The sub-strains are then connected based on the Hamming distance between pairs of sub-strains in the overlapped areas. The full strains are further corrected in the overlapped areas by finding the consensus of SNP fragment matrices from consecutive regions. Strain frequencies are estimated using an expectation-maximization algorithm as in (Eriksson *et al.*, 2008). In the end, CAECseq reconstructs 2 full Zika virus strains with frequencies $77.45\%$ and $22.55\%$, achieving MEC of 357475. TenSQR reconstructs 2 full Zika virus strains with frequencies $72.38\%$ and $27.62\%$, achieving MEC of 365487. PredictHaplo only reconstructs the dominant strain found by CAECseq and TenSQR, achieving MEC of 377364. Note that the runtimes of all the other competing methods exceeded 48 hours and thus the results for those methods are unavailable.