[Reviews · NeurIPS 2020]

Review 1

Summary and Contributions: - This paper presents a method, CAECseq, to recover or identify clusters & consensus sequences from sequential discrete data by combining a convolutional auto-encoder and a clustering layer. As opposed to previous approaches, their approach takes positional information in the input into account while requiring only a third as much computation as the next best, current SoTA approach does. They apply this method to two benchmark task types, haplotype assembly and viral quasispecies reconstruction. At both they consistently perform at or better than the current state SoTA method.

Strengths: - CAECseq consistently performs better than the other methods to which they compared it across both tasks. - CAECseq scales better than the second-best method, GAEseq, by allowing for mini batch gradient descent, thereby reducing runtime, as shown in their first experiment.

Weaknesses: - CAECseq seems similar to GAEseq insofar as it leverages an auto-encoder framework with neural networks layers serving as encoder/decoder. While CAECseq consistently performs better than GAEseq on the two benchmarks, it would be helpful to understand why the authors believe this to be the case. - Given that the CAECseq algorithm involves multiple steps outside of the main optimization loop, it would have been nice to have more experiments testing how CAECseq's performance changes when certain aspects are changed or removed. For example, what happens if the clusters don't get initialized using k-means? How much worse does the method perform without the MEC post-optimization step?

Correctness: - The method as described makes sense.

Clarity: - In general, the paper describes the method clearly but lacks motivation for some of the choices. For example, in section 2.3, where does the KL loss under the p distribution come from? It's stated that it indirectly minimizes the MEC loss while allowing for minibatch optimization, but not explained how. - Minor: In section 2.4, it's not immediately clear that steps 2, 3, and 4 refer to those steps in the __prior__ section.

Relation to Prior Work: - I'm not an expert on this area, but the authors' treatment of prior work seems thorough.

Reproducibility: Yes

Additional Feedback: - The authors shared parameter settings and described the algorithm in enough detail that I believe I could reproduce it. - The authors mention that convolutional layers may perform better due to maintaining relationships between SNPs. A simple experiment to verify this would be to apply the method to a training / test set in which the SNP matrices all have their column indices shuffled consistently. - Have the authors considered slowly increasing gamma (change the weighting between L_r and L_c) as training progresses similar to increasing beta when training a beta-VAE? Perhaps (optimistically) this could remove the need for pre-training on only reconstruction loss, although it would require a different strategy for initializing the cluster layer weights? - What are additional example tasks to which the authors can imagine this method applying? Is there a potential application for this method to GWAS data? Would be nice to have in the conclusion.


Review 2

Summary and Contributions: This paper describes a method for haplotype and viral quasispecies reconstruction. The problem is to recover a small number of long sequences from a large set of short fragments sampled from these sequences. The proposed method applies an autoencoder on one-hot encoded fragments, and recovers the original sequences (haplotypes or quasispecies) by clustering fragments based on the learnt latent representation of the autoencoder. The method is evaluated on the two types of tasks, where is outperforms other methods both in speed and reconstruction accuracy.

Strengths: The reconstruction problem tackled by the submission is important, and the choice of an autoencoder with convolutional layers to build the representation space seems relevant, as CNNs are known to perform well on several learning tasks from biological sequences. The writing is generally clear. The proposed method seems to perform well both in terms of speed and accuracy.

Weaknesses: The main weakness of this submission is the limited methodological novelty of the proposed method. Neither convolutional networks to model biological sequences, and autoencoders for clustering are new. Using this approach to address haplotype reconstruction is new to the best of my knowledge, and seems like a useful contribution, but would be much better suited to a bioinformatics venue. I also found that some aspects could be better justified (although most of the submission is clearly explained). The autoencoders seems to take aligned reads as inputs, but the text only mentions reads, (which could also be unaligned). The fact that Figure 2 shows an empty cell in the middle of the example read is also confusing: is it a gap created during alignement? I was also confused by the clustering objective, which is the KL divergence between the current probabilistic cluster assignment q and a partition p of the reads. However, p is obtained by clustering using q, which seems circular. What is the impact of the gamma parameter controlling the relative importance of the clustering and autoencoder objectives (gamma=0.1 is used in the experiments)? Would the results be very different using only the k-means step for clustering?

Correctness: The claims, method and empirical methodology are correct.

Clarity: The submission is generally well written.

Relation to Prior Work: Related work is correctly summarized and discussed.

Reproducibility: Yes

Additional Feedback: My original score was low because I found the methodological contribution of the submission limited for neurips. The proposed model combines well known elements: autoencoder, clustering in the latent space, one-hot encoding for sequences. It does not introduce any conceptual or algorithmic novelty. However, other reviewers pointed out the fact that the call for paper specifically asked for applications to computational biology. This submission brings a significant improvement to an important problem using machine learning, and provides a good experimental validation. My new score is not larger than 6 because I still think that the relevance to the neurips community is limited (ie, high for members interested in computational biology, lower for the others).


Review 3

Summary and Contributions: This paper describes a neural network architecture for haplotype reconstruction. This is a very well-studied, NP-hard problem for which one neural network-based solution already exists. Whereas the previous work used graph encoding and an explicit decoder into haplotypes, the current work uses convolutions on a one-hot encoding and then uses a "clustering layer" to represent the haplotypes. In practical terms, the main difference is that the new architecture is trained in a minibatch fashion. While this approach only indirectly optimizes the desired loss function, it does allow the method to scale better. Empirical results support the claim that the method improves upon previous work.

Strengths: The problem is very important. The description of the problem is clear enough to be understandable to a typical neurips attendee. The neural net architecture is sensible. The experimental setup is impressive. Careful thought was put into generating realistic simulated data using a variety of state-of-the-art tools. The method is compared against several state-of-the-art techniques. The experimental results are strong. The proposed method strongly outperforms competitors, both in terms of accuracy and running time.

Weaknesses: The primary claim seems to be that the proposed method improves over previous work primarily by using minibacth training. This is summarized by the last sentence of the introduction (line 122-123): "Note, however, that due to aiming to minimize the MEC score directly, GAEseq uses full-batch gradient descent which makes it exceedingly slow and practically infeasible when dealing with large numbers of reads." Other weaknesses are mentioned below. Minor: I don't understand in what sense the low-dimensional embedding is claimed to be "stable" (line 67 and line 191).

Correctness: Yes

Clarity: One of the key methodological innovations here is the inclusion of a clustering layer. This should be explained (along the lines of lines 175-177) when this layer is first introduced in line 55. The MEC score should be defined when it's first mentioned (line 70). The strategy or inferring population size should be briefly explained at line 148. A cite should be given for PReLU (line 168). I could not understand this sentence (line 170-171): "Note that the learned features are restricted to be shorter than haplotypes to avoid learning useless features." The sentence, "The reported results were obtained on test data" (line 207) is insufficiently clear. I am guessing this means that in addition to the data described in the preceding sentence, the authors independently generated test data following a similar protocol to the training data.

Relation to Prior Work: The paper contains an impressive list of prior work in this area, with a phrase describing what each one does. The methods are appropriately grouped. The most closely related work is the graph auteoncoder of Ke and Vikalo. The paper implies that the main difference is that the new method can be trained in a batch fashion. This leads me to wonder why the GAEseq method couldn't be trained using minibatches. The main unanswered question here is why CAECseq outperforms GAEseq. The text seems to imply that this is solely because of the switch to minibatch training, but surely the change from a graph-based representation to one-hot encoding with convolutions is also significant. To me, the failure to more completely discuss the differences between the two methods and to compare intermediate variants is the biggest missing piece of this paper. On a related note, the paper makes several mentions of how the convolutional layers "capture spatial relationship [sic] between SNPs." I assume by "spatial" the authors actually mean positional relationships along the genome (which, I suppose, is technically a 1D space). The claim that this ability "enables the proposed method to distinguish reads obtained from highly similar genomic components" (line 76-77) is not very clear and is not supported by any evidence.

Reproducibility: Yes

Additional Feedback: After reading the feedback and taking into account the other reviews (specifically, the relatively small methodological innovation), I lowered my score by one point.


Review 4

Summary and Contributions: The authors address the problem of haplotype assembly: that is they aim to take as input a set of short sequencing reads and output a set of reconstructed haplotypes that summarize the putative sequences those reads came from. The authors propose a method that first projects sequenced reads to a low-dimensional space and then estimates the probability of the read origin using learned embedded features. Their method is based on a convolutional auto-encoder, which means they first project the reads to a low-dimensional space while maintaining the spatial relationships between SNPs by using the convolutional auto-encoder. They evaluate their method using synthetic data on Solanum Tuberosum data and real data on HIV-1 according to the agreement of reads and reconstruction (minimum error correction, MEC) and agreement with ground truth (reconstruction rate / correct phasing rate, CPR). Their method outperforms others on synthetic MEC; on other metrics, it performs similarly to existing methods.

Strengths: Overall, their approach and idea are interesting, and the results seem promising.

Weaknesses: As noted below, (1) it is not clear what the contribution of their method is, and (2) necessary details are missing from their evaluations. 1_ More detail on the data sets and experiments is required: They test their method on TB and HIV, but for instance, there is no information in the paper that what is the size of the dataset, the source of their dataset, or how they split their dataset into test and train. Also, at some points of their approach in preparing the dataset, they used some restrictions on some parameters but they didn’t mention why they used these numbers. This is concerning, so I think they need to clarify why they used those parameters. For example, for TB they mention that: “Read alignment is performed using BWA-MEM (Li and Durbin, 2009), where the reads with mapping scores lower than 40 are filtered out for quality control." But for HIV they choose different approach: “Reads with mapping score lower than 60 and length shorter than 150 bp are filtered out for quality control". 2_ I am not clear on what the contribution of this work is. The manuscript includes a section stating "Our main contributions are summarized as follows", but this actually just gives a summary of their approach and does not say which parts are novel. My best understanding is as follows: They build off of GAEseq (Ke and Vikalo 2020), which also uses an autoencoder to map reads into a low-dimensional space to facilitate haplotype assembly. I think their contribution over GAEseq is that "GAEseq uses full-batch gradient descent which makes it exceedingly slow and practically infeasible". I don't understand this statement and how CAECseq solves this problem. If this is indeed the contribution of the paper, it needs to be explained in more detail. 3_ As noted in 1.2, there are a large number of existing methods for this task. Why did the authors choose the subset they did to compare against? It is not clear to me that they represent the state of the art.

Correctness: No -- necessary details are missing.

Clarity: Yes, overall.

Relation to Prior Work: No -- see above.

Reproducibility: No

Additional Feedback: Minor notes: - Figure 1 has visual artifacts for me when viewed in Preview on OS X. There are no artifacts for me when viewed in Adobe Acrobat, though. - The use of "genomic components" is nonstandard. I believe "contig", or "haplotype" are more specific terms for what the authors are talking about.

[Author Response · NeurIPS 2020]

We thank the reviewers for their feedback. In the following, we first address questions brought up by multiple reviewers.

**_Q1: Why does CAECseq perform better than GAEseq?_** As mentioned in the paper, there are three reasons: (1)
Convolutional layers allow CAECseq to capture spatial relationships (correlations) between SNPs. (2) Mini-batch
stochastic gradient descent helps CAECseq escape local optima while full gradient descent may cause GAEseq to get
stuck in local optima. (3) Nucleotide representation used by CAECseq (one-hot encoding) means the distances between
nucleotides are symmetric while the representation via integers used by GAEseq leads to unjustified asymmetry.

**_Q2: Where does the KL loss under the p distribution come from? How can it indirectly minimize the MEC loss while_**
**_allowing for minibatch optimization?_** The KL loss, $L_c = \sum_i \sum_j p_{ij} \log \frac{p_{ij}}{q_{ij}}$, is equivalent to categorical cross entropy;
here $[p_{ij}]$ form a $k$-dimensional standard unit vector and $\sum_j q_{ij} = 1$. As explained in the paper, $p_{ij}$ are obtained by
reassigning read origins to minimize the MEC loss at each epoch (using all the reads) while $q_{ij}$ is acquired from the
clustering layer at each iteration (using mini-batch of reads). Therefore, mini-batch optimization enables updating $q_{ij}$ at
each iteration, and minimizing the KL loss indirectly minimizes the MEC loss.

**Reviewer 1** Q3: What happens if the clusters don't get initialized using k-means? We will use the p17 region of HIV-1
as an example. In the paper, we reported that on this region CAECseq achieves the MEC score and CPR of 34036 and
100%, respectively. Without the k-means initialization, these deteriorate to 115134 and 54.2%, respectively. Q4: [To
verify that CAECseq captures spatial relationships between SNPs, run a simple experiment] on a training / test set in
which the SNP matrices all have their column indices shuffled consistently. We did so on a randomly shuffled SNP
fragment matrix of the p17 region in HIV-1; the resulting MEC score and CPR are 206475 and 34.6%, respectively.
Therefore, random shuffling destroyed the spatial relationship between SNPs that the convolutional layers originally
captured. Q5: What are additional example tasks? Is there a potential application for this method to GWAS data? An
additional example task is anomaly detection, e.g., in an application to viral sequence classification. Applications to
GWAS data are certainly possible with appropriately formatted input data and may be a part of our future work.

**Reviewer 2** Q6: The main weakness is the limited methodological novelty of the proposed method. The paper presents
the first ever _deep learning_ architecture (autoencoder with a clustering layer) for a pair of challenging problems in
bioinformatics. The method incorporates domain knowledge in a novel and unique way and enables unprecedented
accuracy. We anticipate it will be very valuable in practice as it outperforms classical approaches by orders of magnitude
and is orders of magnitude faster than the only other existing neural network based method (GAEseq, a shallow
architecture). Q7: The AE seems to take aligned reads as inputs, but the text only mentions reads (which could also be
unaligned). We explicitly state that the reads are mapped to a known reference genome (please see Section 2.1, line
128). Q8: An empty cell in the example read in Fig. 2 is confusing. The empty space models gaps in coverage of
paired-end reads. Q9: What is the impact of the gamma parameter? We tested CAECseq for $\gamma$ varying from 0.01 to
0.99. A large $\gamma$ distorts the feature space by reducing the ability of the convolutional AE to learn salient features of
reads while a small $\gamma$ implies that the model does not put much effort in the reconstruction task. Q10: Would the results
be very different using only the k-means step for clustering? Yes. E.g., on p17 region of HIV-1, $k$-means achieve the
MEC score and CPR of 152464 and 46.6%, while CAECseq on the same task achieves 34036 and 100%, respectively.

**Reviewer 3** Q11: In what sense is the low-dimensional embedding "stable"? We refer to a low-dimensional embedding
as being stable if it helps minimize the MEC score when the clustering layer is employed. Q12: A key methodological
innovations here is the inclusion of a clustering layer. This should be explained [...] A cite should be given for PReLU.
Thanks, we are committed to making these updates! Q13: I could not understand this sentence (line 170-171). A
clarification: if the dimension of the learned features is larger than the length of haplotypes, the auto-encoder only learns
to copy the input (i.e., $f(x) = x$, thus learning non-informative features). Q14: "The reported results were obtained on
test data" (line 207) is insufficiently clear. We tuned the hyper-parameters and validated them on ten simulated tetraploid
datasets, and then applied them to all the datasets in the paper. Therefore, all the datasets in the paper are test data.
We do not split datasets into training, validation and testing parts because such splitting reduces sequencing coverage
and thus reduces reconstruction accuracy. Q15: Why the GAEseq method couldn't be trained using minibatches.
Calculation of the MEC score, which GAEseq aims to directly minimize, requires using all the reads at each iteration.
Q16: "enables the proposed method to distinguish reads obtained from highly similar genomic components" (line
76-77) is not very clear and is not supported by any evidence. We compared the performance of CAECseq with other
SOTA methods on simulated viral quasispecies data with diversity from 1% to 10% in Supplementary Document D.

**Reviewer 4** Please see our answers to questions Q1, Q2 and Q14. Q17: Clarify the choices of parameters. Mapping
quality scores 40, 60 and read length 150 bp are standard in literature and practice of haplotype assembly. For viral data,
higher mapping quality score (60) is needed since the assembly task is generally more challenging. Q18: Why did the
authors choose the subset they did to compare against? Extensive literature review reveals that GAEseq, HapCompass,
H-PoP, AltHap, TenSQR are state-of-the-art methods that outperform other existing techniques in terms of accuracy;
moreover, while other methods are restricted to bi-allelic diploid data, these can handle multi-allelic polyploid data.

[Meta-Review · NeurIPS 2020]

All referees vote for acceptance, and I also fully support this point of view, although the methodological contribution might be somewhat limited. However, it is a very solid application paper.